# DNA: Proximal Policy Optimization with a Dual Network Architecture

**Matthew Aitchison**
The Australian National University
`matthew.aitchison@anu.edu.au`

**Penny Sweetser**
The Australian National University

## Abstract

This paper explores the problem of simultaneously learning a value function and policy in deep actor-critic reinforcement learning models. We find that the common practice of learning these functions jointly is sub-optimal due to an order-of-magnitude difference in noise levels between the two tasks. Instead, we show that learning these tasks independently, but with a constrained distillation phase, significantly improves performance. Furthermore, we find that policy gradient noise levels decrease when using a lower *variance* return estimate. Whereas, value learning noise level decreases with a lower *bias* estimate. Together these insights inform an extension to Proximal Policy Optimization we call *Dual Network Architecture* (DNA), which significantly outperforms its predecessor. DNA also exceeds the performance of the popular Rainbow DQN algorithm on four of the five environments tested, even under more difficult stochastic control settings.

## 1 Introduction

Combining deep neural networks with reinforcement learning has produced impressive results on challenging problems, such as playing Chess [32], Atari games [25], and robotics tasks [31]. However, results have bifurcated between two competing approaches: Q-learning-based approaches that require learning an (action conditioned) value estimate and actor-critic policy gradient (AC-PG, or just PG)[1] approaches that learn both a policy and value estimate. PG offers many theoretical advantages over Q-learning, such as natural support for continuous control problems and the ability to learn stochastic policies. Until recently, PG approaches [24, 31, 11], while strong on continuous control problems, have under-performed Q-learning approaches on vision-based tasks [25, 16, 4], restricting the use of PG in this domain.

This paper aims to close the gap between PG and Q-learning methods by showing that the common practice of jointly learning value and policy with shared features negatively affects the algorithm's performance.[2] We also demonstrate an order-of-magnitude difference in noise levels between these two tasks and argue that this makes these two tasks poorly aligned.

In light of this result, we introduce a new algorithm, based on Proximal Policy Optimization (PPO) [31], called Dual Network Architecture (DNA). We test DNA empirically on a subset of the Arcade Learning Environment (ALE) [6], Atari-5 [2]. Our results show a strong increase in performance on this benchmark compared to both PPO and another dual network model Phasic Policy Gradient (PPG) [11]. Our model also outperforms Rainbow DQN [16] on four of the five games tested, even under more challenging environmental settings.

---

[1]For simplicity, we refer to actor-critic policy gradient as policy gradient, even though some early policy gradient approaches, such as REINFORCE [36], do not learn a value function.

[2]PPO [31], A3C [24], and Muslie [15] all use a joint network when training on vision-based discrete action environments. PPG [11] uses a dual network with an auxiliary task, which we discuss in more detail in Section 2

36th Conference on Neural Information Processing Systems (NeurIPS 2022).

We summarize our contributions. First, we provide empirical results showing an order-of-magnitude difference in the noise scale between the policy gradient and the value loss gradient. Second, we give evidence for the benefit of using low *bias* value estimates for value learning, but low *variance* estimates for advantage estimates. Finally, we introduce and justify our dual network constrained distillation algorithm DNA with empirical results on ALE.

## 2  Preliminaries and Related Work

**Proximal Policy Optimization**   Our algorithm builds on the well-established Proximal Policy Optimization (PPO) algorithm [31]. PPO is a policy gradient, reinforcement learning algorithm, with many of the advantages of its predecessor Trust Region Policy Optimization [29], while being much simpler to implement. Various aspects of PPO have been studied in recent years, such as the impact of implementation choices [12, 19], the ability to generalize [11, 27], and performance under multi-agent settings [37]. There has also been a growing understanding that the value and policy learning tasks involved in actor-critic models like PPO have important asymmetries [27, 11]. Our work builds on this by investigating how differences in value and policy noise levels can be accommodated to improve PPO's performance on video-based discrete action tasks.

**TD($\lambda$) Return Estimation**   Policy gradient algorithms often make use of a value estimate as a baseline [24], as well as for estimating the value of truncated trajectories [9]. To better facilitate control over the noise levels for policy and value learning, our work makes use of two different return estimations, both using TD($\lambda$) [33]. Given value estimates $V(\cdot)$ from the value network we define the n-step value estimate for some state $s_t$ taken at time $t$, and their exponentially weighted sum as

$$\text{NSTEP}^{(\gamma,k)}(s_t) := \sum_{i=0}^{k-1} \gamma^i r_{t+i} + \gamma^k V(s_{t+k}), \tag{1}$$

$$\text{TD}^{(\gamma,\lambda)}(s_t) := (1-\lambda) \sum_{k=1}^{\infty} \lambda^{k-1} \text{NSTEP}^{(\gamma,k)}(s_t). \tag{2}$$

For values of $\lambda$ close to 1, more weight is assigned to longer n-step return estimates, and less to shorter ones. There has been a long-standing belief that shorter n-step returns generate more biased estimates, whereas longer n-step estimates, due to summing over many stochastic rewards, have higher variance.[3] For a more thorough discussion on this topic see [20].

**Noise Scale.**   Noise scale is a measure of noise-to-signal in stochastic gradient descent (SGD). While SGD provides unbiased gradient estimates, these estimates are typically very noisy. Each sampled gradient estimate $\hat{G}$ can be thought of as a sum of the true gradient $\vec{G}$ and some noise vector $\vec{\sigma}$. A method for estimating the ratio of the magnitude of this implied noise vector to the magnitude of the true gradient was proposed by [23] who show that their efficient-to-calculate *simple noise scale* is a good match for the noise scale. They also show that the noise scale provides useful information about the choice of mini-batch size to use when estimating gradients.

**Phasic Policy Gradient.**   Most similar to our work is Phasic Policy Gradient (PPG) [11]. Like DNA, PPG has three distinct phases during training and uses two independent networks. However, there are several important distinctions in our work. First, we forgo the large replay buffer, relying instead on learning entirely from recent experience. This significantly reduces the memory requirements of our algorithm. Second, we make use of a distillation phase rather than an auxiliary phase.[4] Finally, our work reduces gradient noise by using two different return estimators calibrated for the characteristics of each task. We assess the impact of these differences on the performance of the agent in Section 6.

---

[3]While there are cases (i.e. where rewards are negatively correlated) where this is not the case, empirical experiments confirm the intuition that longer n-step estimates generally have higher variance and lower bias.

[4]PPG refer to their third phase as a distillation phase, however, because the update trains both the policy and value networks on value estimates generated from rollouts, it is better thought of as an auxiliary task. We discuss this in more detail in Section 4.3

# 3 The Noise Properties of Value Learning and Policy Gradient

Here we examine the noise properties of the policy gradient and value learning tasks and show that the policy gradient has a much higher noise level than value learning. This implies different challenges in the optimization problem. Specifically, that policy should be learned with a much larger mini-batch size than value learning and that return estimates should be adapted appropriately.

## 3.1 Motivating Example

To motivate our investigation into noise levels, we consider the task of learning two independent functions $F_1(x) := \sin(5x) + \mathrm{N}(0, \sigma_1^2)$ and $F_2(x) := \cos(5x) + \mathrm{N}(0, \sigma_2^2)$, where $\mathrm{N}(\mu, \sigma^2)$ is Gaussian noise with mean $\mu$ and standard deviation $\sigma$, over the domain $[-\pi, \pi]$. We fix $\sigma_2 = 1$, and vary $\sigma_1$ on a log scale from $0.1$ to $100$. We trained two 3-layer multi-layer perceptions (MLPs) on this problem.[5] The first was a joint model, using a first hidden layer of 1024 units and a second hidden layer of 2048 units, followed by two output heads. Our second model was a dual network consisting of two independent MLPs with 1024 units on each hidden layer.[6] All models used ReLU activations in between linear layers, and error was measured as mean-squared-error (MSE) between the predicted value and the *noise free* true value. The results are presented in Figure 1. At low noise levels, the tasks did not interfere, but as the noise of the first task increased, performance on the second task eventually degraded for the joint network, but not the dual network. While simple, this experiment demonstrates the impact one noisy task can have on another when learned jointly by a deep neural network.

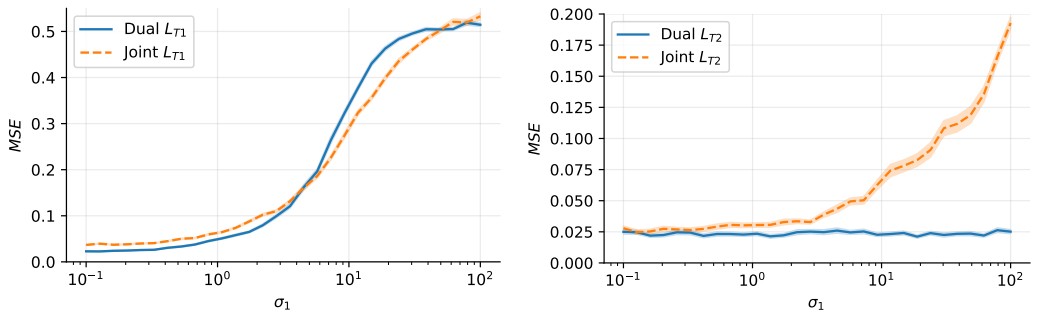

Figure 1: A depiction of the problem of destructive interference on a toy problem. $L_{T1}$ and $L_{T2}$ refer to the mean-squared-error on the first and second task respectively. Shading indicates standard error over 100 seeds.

## 3.2 Noise Scale in Reinforcement Learning

To assess the differences in noise levels in our experiments, we measure the *gradient noise scale* of the policy gradient and value loss gradient. We did this by calculating an estimate of the noise scale, developed by [23] called *simple noise scale* defined as

$$\mathcal{B}_{\text{simple}} := \frac{\mathrm{tr}(\Sigma)}{|\vec{G}|^2}, \tag{3}$$

where $\Sigma$ is the gradient covariant matrix, and $\vec{G}$ is the true gradient. For convenience we also use the notation $\sigma := \sqrt{\mathcal{B}_{\text{simple}}}$, which can interpreted as the ratio of the length of the implied noise vector to the length of the true gradient vector. That is, a kind of noise-to-signal ratio. As [23] have shown, a low bias estimate of $\mathcal{B}_{\text{simple}}$ can be found efficiently by generating gradient estimates $\vec{G}_{B_{\text{small}}}$ using a small mini-batch of size $B_{\text{small}}$ as well as the gradient $\vec{G}_{B_{\text{big}}}$ using a large mini-batch of size $B_{\text{big}}$, then calculating

---

[5]It is worth noting that we did not find the same result when using shallow networks. That is, dual and joint models performed similarly when using a 2-layer MLP.

[6]This number of hidden units was chosen so that both models had the same number of parameters.

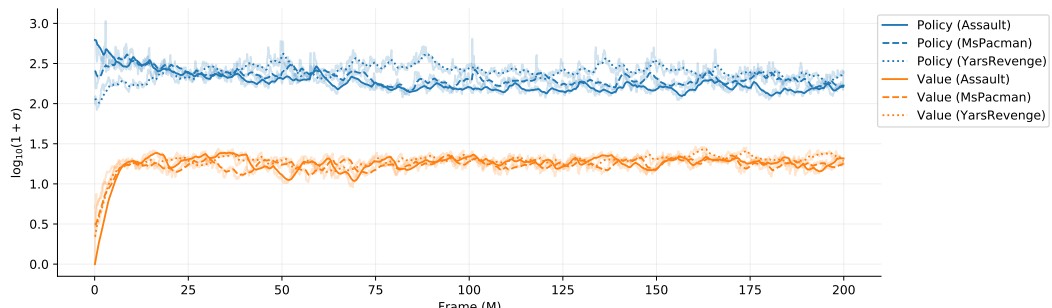

Figure 2: Noise scale for the two learning objectives. For clarity all values are further smoothed using an exponential rolling average (with non-smoothed values presented in a lighter shade). Smoothing may introduce bias, as while $|\mathcal{G}|^2$ and $\mathcal{S}$ are unbiased estimators, their ratio may not be.

$$|\mathcal{G}|^2 := \frac{1}{B_{\text{big}} - B_{\text{small}}}(B_{\text{big}}|\vec{G}_{B_{\text{big}}}|^2 - B_{\text{small}}|\vec{G}_{B_{\text{small}}}|^2) \tag{4}$$

$$\mathcal{S} := \frac{1}{1/B_{\text{small}} - 1/B_{\text{big}}}(|\vec{G}_{B_{\text{small}}}|^2 - |\vec{G}_{B_{\text{big}}}|^2), \tag{5}$$

where $\mathbb{E}[|\mathcal{G}|^2] = |\vec{G}|^2$ and $\mathbb{E}[\mathcal{S}] = \text{tr}(\Sigma)$.

To evaluate the noise scale of value learning and policy gradient we trained our dual network model (described fully in section 4) on the three Atari games from the Atari-3 validation set [2]. Like [23], we found it necessary to smooth out noise by maintaining an exponential moving average of $|\vec{G}_{B_{\text{big}}}|^2$. We used $B_{\text{small}} = 32$ and $B_{\text{big}} = 16,384$ in all our experiments. Pseudocode for this procedure is given in Appendix N.

The results are presented in Figure 2. We found that policy noise ($\sigma_\pi$) reduced by about half during training and that value noise ($\sigma_v$) remained constant. Both noise levels varied very little between the three environments. A large variation was observed between the value loss and policy gradient, consistent between environments. The noise levels, measured at the end of training and averaged over all three environments, were found to be $\sigma_\pi = 220.5$ and $\sigma_V = 17.5$, representing a 12.6 times difference.

## 4 Dual Network Architecture

Based on our experimental results in Section 3, we propose an architecture which takes into account the large difference in noise levels between the two tasks. This architecture consists of three improvements to PPO. First, to reduce negative interference from the noisy policy gradient, policy and value should be learned by independent networks with different hyperparameters (e.g. training epochs and mini-batch size). Second, the variance/bias trade-off in the return estimations should be calibrated to each task. Finally, a constrained distillation phase should be applied to take advantage of any constructive interference between the two tasks.[7]

### 4.1 Independent Networks

Important to DNA is the use of a dual network architecture.[8] Not only does this setup allow for learning of the value and policy without destructive interference, it also enables a specialized set of hyperparameters to be calibrated for the distinct tasks of policy and value learning. Specifically, the

---

[7]The source code used to generate the results in this paper is provided in the supplementary material. We also provide an implementation of our algorithm at `https://github.com/maitchison/PPO/tree/DNA`.

[8]By dual network, we mean two independent networks. Not to be confused with a dual-head network (which we refer to as a joint network), or with dualing networks [35].

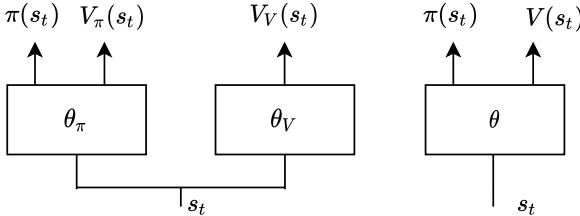

Figure 3: Architecture of DNA (left) compared to a single network setup (right).

calibration of the mini-batch size, which it has been suggested is best set roughly proportional to $\sigma$ [23]. The policy network outputs a policy $\pi$ and a value estimate $V_\pi$, whereas the value network only outputs a value $V_V$, as depicted in Figure 3.

## 4.2 Decoupled Return Estimation

To better facilitate control over the noise levels for policy and value learning, DNA makes use of two different return estimations, both using TD($\lambda$) [33] as follows

$$V_{\text{targ}}(s_t) := \text{TD}^{(\gamma, \lambda_V)}(s_t), \tag{6}$$

$$V_{\text{adv}}(s_t) := \text{TD}^{(\gamma, \lambda_\pi)}(s_t), \tag{7}$$

where $\lambda_V$ and $\lambda_\pi$ are hyperparameters controlling the variance / bias trade-off of each estimate. The $V_{\text{targ}}$ estimates are used as targets for training the value function, whereas $V_{\text{adv}}$ are used for advantage estimates used for policy gradient estimates given by

$$\hat{A}_t := V_{\text{adv}}(s_t) - V_V(s_t). \tag{8}$$

This formulation of the advantages is equivalent to the general advantage estimation (GAE) [30], that is, for any $\lambda \in [0..1], \gamma \in [0..1)$ we have,[9]

$$\hat{A}_t^{\text{GAE}(\gamma, \lambda)} = \text{TD}^{(\gamma, \lambda)}(s_t) - V_V(s_t). \tag{9}$$

Because of this, it is common when using PPO to generate value targets, $V_{\text{targ}}$, for the value function by adding the network's value estimates to the (unnormalized) advantages, implicitly settings $\lambda_V = \lambda_\pi$.[10] We hypothesize that due to bootstrapping, estimates used for $V_{\text{targ}}$ would benefit from being *low bias*, conversely, because estimates used for advantage estimation contribute to the noise of the (already very noisy) policy gradient, they would benefit from being *low variance*. That is, the bootstrapping process may cause symmetrically distributed error, but not biased error, to cancel out. This suggests setting $\lambda_\pi < \lambda_V$, a thought which we test empirically in section 6.2.

## 4.3 Distillation

Because value learning is a simpler task in terms of noise level, it makes sense to try to transfer some of the knowledge learned by the value network to the policy network. If the value network learns to identify important features in the environment, we would like the policy network to identify those features also. To facilitate this, we employ a constrained distillation update [17]. Distillation between two identical networks has the property that the student's target function (since an identical network generated it) is guaranteed to be in the model's solution space. Also, because the distillation targets are deterministic,[11] the process is also very low noise, which we verify in Appendix D.

---

[9] A proof for this claim is provided in Appendix E

[10] For example, see line 65 of the popular baselines implementation of PPO. https://github.com/openai/baselines/blob/ea25b9e8b234e6ee1bca43083f8f3cf974143998/baselines/ppo2/runner.py

[11] That is to say, $V_{\text{targ}}(s_t)$ are samples drawn from the random variable $R(s_t)$, where $R$ is a (noisy) return estimator, whereas $V(s_t)$ is a deterministic function that produces an estimate of $\mathbb{E}[R(s_t)]$.

We contrast this with the auxiliary update of PPG which trains both the value function and the policy on an auxiliary task. This auxiliary task equates to learning $\frac{1}{N}\sum_{i=0}^{N-1}V_{\text{targ}}^{\pi-i}$ where $V_{\text{targ}}^{\pi-i}$ is a value estimate for the policy from $i$ updates prior. For quickly changing policies, and large $N$, these updates may cause tension with the value estimates of the current policy.[12]

We use $V_V(s)$ as the distillation targets, with the input states $s$ being taken from the current rollout. However, unlike policy and value training, distillation state inputs need not be on-policy. We explored other distillation targets, covered in Appendix J, but found $V_V(s)$ to be the best of those tried. Distillation is performed, like PPG's auxiliary task, using mean squared error, and under a soft constraint on the policy network's policy, specifically

$$L_t^D(\theta) := \hat{\mathbb{E}}_t[(V_\pi(s_t) - V_V(s_t))^2] + \beta \cdot \hat{\mathbb{E}}_t[\text{KL}(\pi_{\text{old}}(\cdot|s_t), \pi(\cdot|s_t))] \qquad (10)$$

where $\beta$ is the policy constraint coefficient, and $\pi_{\text{old}}$ is a copy of the policy before the distillation update. During updates gradients are only propagated through the policy network, and not the value network.

## 4.4 Training

Like Phasic Policy Gradient (PPG) [11], DNA splits training into three distinct phases (policy, value, and distil), but unlike PPG, rather than using a large replay buffer, we perform all updates on-policy on the current batch of rollout data. Each of the three phases optimizes a single objective, for some number of epochs, using its own optimizer, with a unique set of hyperparameters. The optimization objective for the policy phase of DNA is the clipped surrogate object from PPO [31] including the entropy bonus

$$L_t^{\text{CLIP}} := \hat{\mathbb{E}}_t\left[\min(\rho_t(\theta)\hat{A}_t, \text{clip}(\rho_t(\theta), 1-\epsilon, 1+\epsilon)\hat{A}_t) + c_{\text{eb}} \cdot \text{S}[\pi(s_t)]\right] \qquad (11)$$

where S is the entropy in nats, $\rho_t$ is the ratio $\frac{\pi(a_t|s_t)}{\pi_{\text{old}}(a_t|s_t)}$ at time $t$, $\epsilon$ is the clipping coefficient, and $c_{\text{eb}}$ is the entropy bonus coefficient. For the value phase, we we use the squared-error value loss,

$$L_t^{VF} := \hat{\mathbb{E}}_t\left[(V_V(s_t) - V_{\text{targ}}(s_t))^2\right]. \qquad (12)$$

In summary, the DNA algorithm separates the tasks of policy learning and value learning into a network to handle the high noise of policy learning and a separate network to handle the lower noise task of value learning. Knowledge from the value learning network is transferred to the policy network through a separate constrained distillation phase, which allows for constructive interference between the learning tasks while minimizing the destructive. We formalize the algorithm as follows.

---

[12]A better way to do auxiliary tasks would be to train a separate value network output head, dedicated to the auxiliary task. This way the agent could learn the value of the current policy, along with the moving average value. We believe, however, that the better solution to overfitting the value function to recent data is simply to train less. An idea which we discuss further in Section 7.

---

**Algorithm 1** Proximal Policy Optimization with Dual Network Architecture

---
1: **procedure** PPO-DNA
2:     **Input** $N \in \mathbb{Z}^+$ rollout horizon
3:     **Input** $A \in \mathbb{Z}^+$ number of agents
4:     **Input** $\pi$ the initial policy.
5:     **for** $i = 1$ to ... **do**
6:         **for** $a = 1$ to $A$ **do**
7:             Run policy $\pi$ in environment $a$ for $N$ timesteps
8:         Compute $V_{\text{targ}} \leftarrow \text{TD}^{(\gamma, \lambda_V)}$
9:         Compute $V_{\text{adv}} \leftarrow \text{TD}^{(\gamma, \lambda_\pi)}$
10:       Compute $\hat{A} \leftarrow V_{\text{adv}} - V_V$
11:       **for** $j = 1$ to $E_\pi$ **do**
12:           Optimize $L^{CLIP}$ wrt $\theta_\pi$
13:       **for** $j = 1$ to $E_V$ **do**
14:           Optimize $L^{VF}$ wrt $\theta_V$
15:       $\pi_{\text{old}} \leftarrow \pi$
16:       **for** $j = 1$ to $E_D$ **do**
17:           Optimize $L^D$ wrt $\theta_\pi$
18:     **Output** $\pi$

---

# 5 Evaluation

To evaluate our algorithm's performance, we used the Atari-5 benchmark [2]. Scores in Atari-5 are generated using a weighted geometric average over five specific games and produce results that correlate well with the median score if all 57-games had been evaluated. This allowed us to perform multiple seeded runs and defined a clear training and test split between the games. In all cases, we fit hyperparameters to the 3-game validation set and only used the 5-game test set for final evaluations.

We opted for the more difficult stochastic ALE settings recommended as best practice by [22]. However, to better understand our results in the context of prior work, we also provide results under the simpler deterministic settings in Appendix F, and additionally provide for reference a single evaluation on the full 57-game set in Appendix M. Unless otherwise specified, agents were scored according to their average performance over the previously completed 100-episodes at the end of training.

A coarse hyperparameter sweep found initial hyperparameters for our model on the Atari-3 validation set. Notably, we found the optimal mini-batch size for value and distillation to be the minimum tested (256), while the optimal mini-batch size for policy was the largest tested (2048). For optimization, we used Adam [21], over the standard 200 million frames.[13] Full hyperparameter details for our experiments are given in Appendix B.

In order to understand the impact of the hyperparameters introduced by our algorithm, namely $E_\pi, E_V, E_D$ and $\lambda_\pi, \lambda_V$, we performed the following experiments on the Atari-3 validation set. We started by setting $E_\pi = E_V = E_D = 2$, then searched over $E_V \in [1, 2, 3, 4]$, and selected the best $E_V$. We then searched over $E_\pi$ and $E_D$ while keeping the other two parameters constant. We also performed a similar experiment on the impact of $\lambda_V$ and $\lambda_\pi$ by setting $\lambda_V$ to 0.95 then sweeping across $\lambda_\pi \in [0.6, 0.8, 0.9, 0.95, 0.975]$, then setting $\lambda_V$ to the best $\lambda_\pi$ and repeating. This process allowed us to verify if the best settings occur at $\lambda_V = \lambda_\pi$ or at $\lambda_V \neq \lambda_\pi$. During our $\lambda$ tuning, we also recorded noise levels for one of the seeds, the results of which we discuss in Section 6.2.

We evaluated our algorithm DNA, against PPO, and PPG. All models used the 'NatureCNN' encoder from [25]. However, because DNA and PPG use two networks and thus twice the parameters, we doubled the number of channels of the PPO encoder which we refer to as PPO (2x) and which is very similar to the encoder used by [4]. All models have have approximately 3.5 million parameters in total. For fairness we repeated the same sweep on PPO for the number of training epochs ($E_{\text{PPO}}$) and $\lambda$ used for GAE ($\lambda_{\text{PPO}}$). To verify that our settings did not inadvertently degrade the performance of

---

[13]That is 50 million environment interactions.

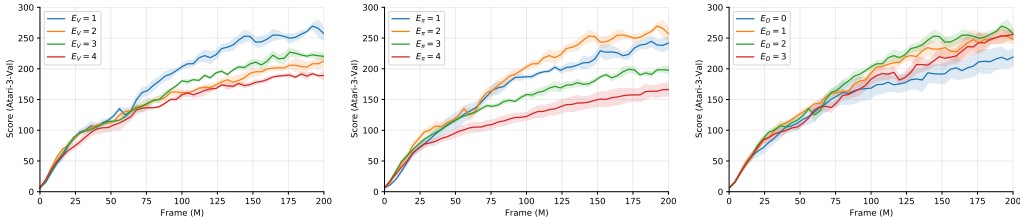

Figure 4: Training curves for DNA with various epochs. The DNA algorithm is most sensitive to the number of policy epochs, and is optimal on our validation set with $E_\pi = 2$. Applying policy updates 3 or 4 times significantly reduces the agents performance.

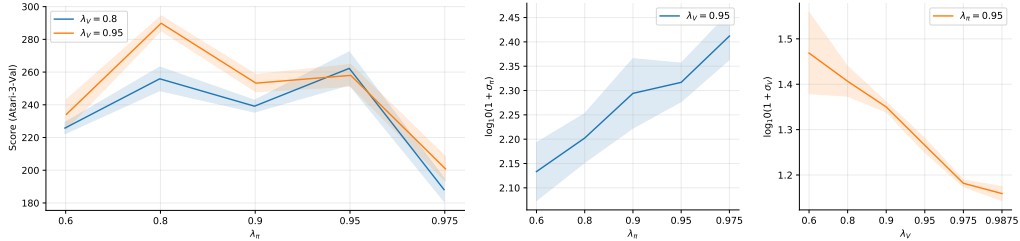

Figure 5: Performance of DNA across a range of $\lambda$ values for return estimations. Left: The optimal settings for $\lambda_V$ and $\lambda_\pi$ differed. Shading indicates standard error over 5 seeds. Mid/right: Noise levels from a single seed, $\sigma_\pi$ increased with $\lambda_\pi$ where as $\sigma_V$ decreased with $\lambda_V$. Shading indicates standard deviation over the three environments.

PPO we also tested against the settings used by [31] and refer to this as PPO (original). These results are included in the ablation study in Appendix C.

## 6 Main Experimental Results

In this section we present the results of our main study comparing DNA to PPO and PPG on Atari-5. We also include supplementary experiments on MuJoCo [34] and Procgen [10] in Appendix K, and L respectively.

### 6.1 Impact of Training Epochs

We start by investigating the impact of the $E_\pi$, $E_V$, and $E_D$ hyperparameters. We found that using *less* than three policy epochs for DNA dramatically increased the performance of the agent while also decreasing the computation required to train the agent (Figure 4). We also found that DNA was robust to the choice of value and distillation epochs but that overtraining on value ($E_V = 4$) marginally decreased performance. This was not the case for distillation updates. The distillation phase demonstrated some benefit as indicated by the $E_D = 0$ results under-perform the others. We selected $E_\pi = 2, E_V = 1, E_D = 2$ as the optimal settings and used these for the remainder of the experiments. These settings require a total of four updates for the policy network, but only one update for the value network (we discuss this further in Section 7). The training curves also suggest DNA may benefit from training beyond the standard 200 million frames.

### 6.2 Return Estimation

We found that the choice of $\lambda_\pi$ made a large difference to the performance of our agent. As was hypothesised the optimal settings for $\lambda_\pi$ differed to that of $\lambda_V$ with $\lambda_\pi$ preferring a lower variance, higher bias value of 0.8, while the value targets, $\lambda_V$ preferred a lower bias, higher variance setting of 0.95 (Figure 5). Notable is that the non-homogeneous setting $\lambda_V = 0.95, \lambda_\pi = 0.8$ outperformed both homogeneous choices $\lambda_\pi = \lambda_V = 0.8$ and $\lambda_\pi = \lambda_V = 0.95$. We found, as expected, that policy gradient noise can be reduced by selecting a low value for $\lambda_\pi$. However, against our expectations, setting $\lambda_V$ lower actually increased, not decreased the noise level of the value learning task. Once we

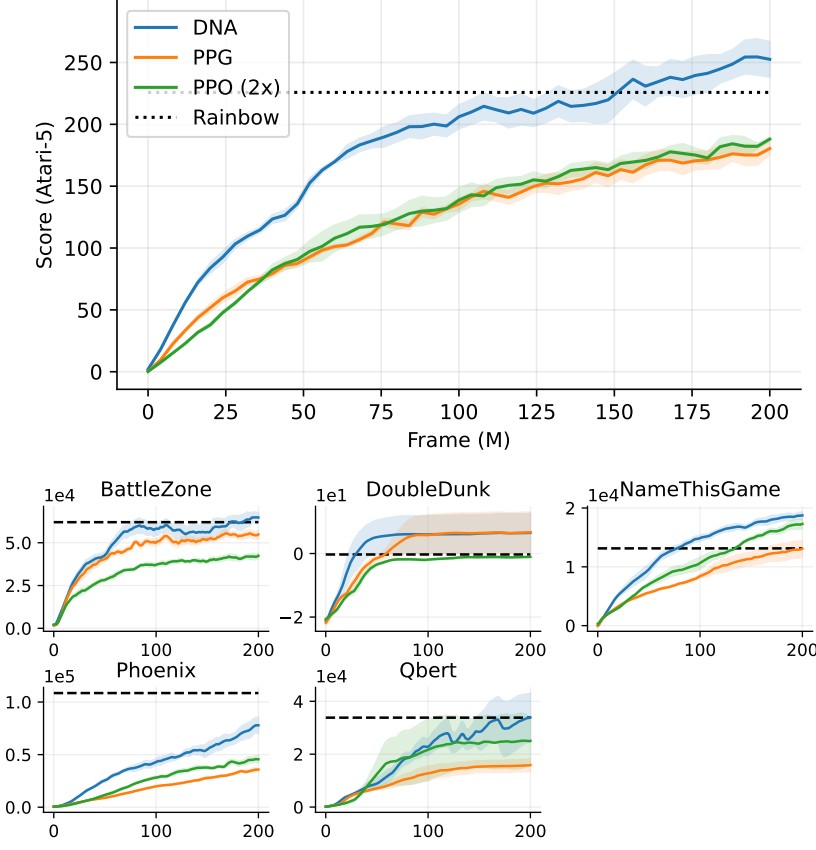

Figure 6: Top: Results on the Atari-5 benchmark. DNA outperforms PPO by a wide margin, and even exceeds Rainbow DQN despite the more difficult settings used in our experiment. Bottom: Individual training curves for each of the five games.

discovered this we ran addition experiments for a broader choice of $\lambda_V$, and include the results in Figure 5.

### 6.3  Comparison to PPO and PPG

We found DNA to outperform both PPO and PPG on the Atari-5 dataset by a wide margin (Figure 6). We give Atari-5 scores, as well as training plots for each individual game. We also include references for Rainbow DQN, although we note that these scores were generated under the simpler deterministic settings.

## 7  Discussion

**Reduced Training as an Alternative to Experience Replay**    We found that *less* training, both on policy and value updates, typically improves our agent's performance. One explanation for this could be, as [11] have suggested, that in the case of value learning, this could be due to overfitting to recent experience.[14] In terms of policy updates, where the largest performance degradation was observed, this could be due to increased training epochs causing a large change in the policy, which may suggest a better constraint than PPO's clipping is required.[15] That being said, limiting the training to one or two epochs provides straightforward solution to the problem while having the additional benefit of decreasing the computational resources required to train an agent.

---

[14]This is plausible, especially with high variance returns.

[15]It is worth noting that PPO does clip the entropy bonus, and therefore updates with a non-zero entropy bonus could make arbitrarily large changes to the policy regardless of the choice of $\epsilon$.

**Choice of** $\lambda$    Our results give evidence for the benefit of selecting $\lambda_\pi < \lambda_V$. As expected, picking *lower* values of $\lambda_\pi$ reduces the policy gradient noise. However, counterintuitively, noise on value learning reduces with *higher* values of $\lambda_V$. That is, policy learning prefers low *variance*, but value learning prefers low *bias*. This result may explain why clipping, which introduces bias, performs poorly on value learning but quite well on policy learning.[16] Our results also suggest that an even higher choice of $\lambda_V$ than the $0.95$ we used in our experiments may be appropriate.

**Update-to-Data Ratio**    Our work also raises an interesting question about the update-to-data (UTD) ratio differences between value and policy learning. We found DNA to work best with only a single value network update but four policy network updates. This asymmetry could be due to the extremely high noise levels for policy updates. The single value update contrasts with other work where much higher value learning UTD ratios was found to be preferred [1, 8].

**Limitations**    Our paper's primary focus was improving PG results on discrete action vision-based problems, as this is where PG has traditionally unperformed Deep Q-learning. However, we do also provide an initial look into continuous control problems on the MuJoCo [34] dataset in Appendix K.

**Broader Impact**    In our work, we have shown that deep actor-critic reinforcement learning models can effectively solve complex vision-based, discrete-action environments, even without the use of large replay buffers. Surprisingly this can be achieved using *less* not more policy and value network updates. Because Q-learning approaches generally produce deterministic policies, algorithms based on them may 'lock in' to a decision based on minimal differences in outcome. This is not always the case with stochastic policies, which are able to randomly split over actions of roughly equal quality, which may lead to fairer outcomes. We do not foresee any direct negative societal impacts from this research.

**Future work**    Our algorithm is already highly competitive with Rainbow DQN, and could likely be extended with orthogonal improvements such as adding recurrence [14], distributional value learning [5], improved feature encoders [13], better exploration bonuses [7], and making use of a learned model [28].

## 8    Conclusion

In this paper, we have highlighted noise level as a key difference between policy and value learning. We have introduced an algorithm that accounts for this order-of-magnitude difference by limiting *negative* interference through the training of two independent networks for value and policy but retaining *constructive* interference through a constrained distillation process. Furthermore, we have shown that the variance/bias tradeoff differs for value learning and policy gradient and that return estimation should cater for this. Together these changes result in a novel algorithm, DNA, that outperforms its predecessor PPO, and even surpasses the popular Q-learning approach Rainbow DQN, while under more challenging environmental settings.

## Acknowledgments and Disclosure of Funding

This research was supported by an Australian Government Research Training Program (RTP) Scholarship.

---

[16]By value clipping, we mean the value trust-region suggested by [30], but which has been shown hurt performance by [3].

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
