# OpenReview forum: "DNA: Proximal Policy Optimization with a Dual Network Architecture"
_NeurIPS.cc/2022/Conference — NeurIPS 2022 Accept_

### Official Review · Reviewer_Knd8 · 2022-06-19

**Rating:** 7
**Confidence:** 3
**Soundness:** 3 good
**Presentation:** 2 fair
**Contribution:** 2 fair

**Summary:**

The paper identifies that the noise level between learning the value function and the policy in reinforcement learning can have an order of magnitude difference. The paper thus proposes to learn these two tasks independently followed by a constrained distillation phase. The paper also argue that the policy gradient noise level can be reduced by using a lower variance return estimate. On the other hand, a lower bias estimate should be used to reduce the value learning noise level.


**Questions:**

Can the authors also reconcile the results presented in the paper with this paper https://arxiv.org/abs/2101.05982? This paper seems to suggest that we should use a high update to data ratio, whereas the current paper seems to suggest that we should perform fewer updates.


**Limitations:**

Yes

**Strengths And Weaknesses:**

Strength

The paper demonstrates the issue of jointly learning two tasks with different noise level in a simple motivating example. The motivating example clearly illustrates destructive interference and provides a great starting point for the discussion in the remainder of the paper.

The paper introduces additional modification on top of PPO, motivated by the previously identified issues, that leads to significant performance Improvement.

Weakness

I would have liked to see a discussion on how the reader should interpret equation 4 and 5 in the paper. Also, How does the y-axis of Figure 2 relate to the left hand side of equations 4 and 5? This is not clearly explained in the paper if I am not mistaken. It would have been helpful if the authors include pseudocode on how the y-axis of Figure 2 is calculated.

Also, the noise scale as explained in equation 3 does not seem to take into account the directions of the true gradient. Is that true? If that is true then the quantity used to compute the noise scale seems a bit odd to me because shouldn't we take into account the direction of the true gradient and the direction of the noise vector as well?

The paper also introduces a new benchmark. Even if an anonymous copy is included, I would have liked to see a self-contained description in the main paper on why results on the new benchmark are meaningful or more meaningful compared to the previous version of the arcade learning environment benchmark.

Can the authors also please elaborate on the discussion at the end of section 4.2 – why is it that because of bootstrapping, estimate used for V Target benefit from being low bias?

I found the exposition in the experimental results section a bit unclear. If the paper is about noise level then why is it that the first experimental result presented has to do with the number of training epochs?

---

> ### Author Response · Authors · 2022-07-31
> **Reponse to Questions**
>
> Thank you for your review. I will try to address each point below.
>
> 1. Noise scale - We were unsure how much detail to go into regarding the noise scale. In the end, we chose to simply refer readers to McCandlish's paper [20], as they do a great job of explaining it. However, we make explicit our interpretation of the ratio between |G|^2 and S, which is the signal to noise and is discussed in the Preliminaries section.
>
> 2. Sigma - The plot in figure 2 is log_10(1+\sigma), where sigma is sqrt(|G|^2/S) and is defined in section 3.2. We would be happy to make this more explicit in the text.
>
> 3. Noise and the direction of the gradient - This is a very interesting point. In their paper McCandlish et. al [20]; provide a noise estimate that does include the direction, which they call B_noise. We chose to use their simplified approximation B_simple, which assumes the Hessian is a scalar multiple of the identity matrix, and which simplifies down to requiring just |G| instead of G. See Page 7 of their paper for more details. Practically speaking, calculating the Hessian was not an option.
>
> 4. Self-contained description of the Atari-5 - Yes, we agree and have now added a paragraph describing and giving justification for Atari-5. Because Atari-5 is new, we also include full results on all 57 Atari games and now also have results on ProcGen and MuJoCo. We hope this will elevate the issue of introducing a new algorithm and a new benchmark at the same time.
>
> 5. Bias and bootstrapping - We agree this was not very clear and have modified the text to explain our intuitive reasoning better here. The idea is that if a value function uses itself to update, then variance errors will average out over time, but bias errors will not. This, of course, may not be true, which is why we tested it empirically.
>
> 6. Preliminary experiments - Regarding the first experiment. Our algorithm introduces several new hyperparameters, and the first experiments are to establish what values should be selected. A key feature of DNA is the ability to adapt individual settings to the policy and value learning tasks. Noise influences the optimal minibatch size, and optimal minibatch generally influences the number of epochs (i.e. larger mini batches typically want more epochs). That is why we needed to check these values. We appreciate the feedback and have adapted our wording to make this point more clear.
>
> Questions
>
> UTD Paper - Great question. We're glad you referenced this paper. We had not seen it before, and it seems highly relevant, given our results, especially now that we have included supplementary results on MuJoCo. We have now included a paragraph discussing it in our paper. One of our more surprising results is that a *low* UTD is critical for PPO and that PPO performs quite well on Atari with just one update. We can only speculate, as others have, that this is due to overfitting (which we discuss briefly in our paper). However, in our additional experiments on Mujoco (included in an updated supplementary), we also found, like the referenced paper, that *high* UTD was critical for MuJoCo. We are not sure why this is the case for Mujoco but not for Atari. An investigation into why atari wants UTD=1-2 and MuJoCo wants 20-40 would be a very interesting future research direction, and we have now included a note regarding that under future work.
>
> We hope these changes have addressed your concerns.

---

> > ### Comment · Reviewer_Knd8 · 2022-08-07
> > **Dear authors**
> >
> > Regarding Eq 4, 5 and the discussion in 3.2, it is still unclear to me how Eq 4, 5 relates to Eq 3. I understand Eq 4, 5 are calculated to obtain an unbiased estimate of the LHS of Eq 3, but how do you obtain this estimate after calculating Eq 4 and 5? I think if you include pseudo-code of the entire procedure to calculate the gradient noise scale, it would be quite helpful. Sorry if I miss this.
> >
> > Regarding the figures, I think more can be done to help the reader interpret the take-aways. The text in the figures are generally very small, and coupled with my previous point, makes interpreting the figures quite non-trivial.
> >
> > Regarding referring to [20], I think it would be helpful if the paper is more self-contained. It seems okay to me if you include more detailed discussion to make the paper self-contained in the Appendix, including what you said in the rebuttal.
> >
> > Can the authors please make the edits to the paper in a different color so it is easier to tell what the changes are?
> >
> > If these changes are addressed, I am happy to increase my score to 7, mainly because I think the analyses regarding noise scale is novel in the context of RL and is a re-usable technique in future algorithmic development.

---

> > > ### Author Response · Authors · 2022-08-08
> > > **Additional Appendix on Noise Scale**
> > >
> > > We have now added Appendix H, which gives Pseudocode for how the $\sigma$ values were calculated for Figure 2, along with an expanded discussion on the topic. We hope this clarifies the exact process of going from gradients to an estimate of the noise scale. As is sometimes the case, we were happy to discover a slight improvement in our algorithm while formalizing this procedure, which we have also noted in the appendix and will use in future work.
> > >
> > > Regarding the size of the plots, we agree. Where possible, we have adjusted the figures' size to make them more readable but are quite limited by the 9-page limit. We would be happy to make use of the 10th page available after acceptance to resolve this issue by increasing their size further. We will also use the extra page to add short statements to the captions to highlight the relevant takeaways, as we agree this would also improve the paper.
> > >
> > > We have now updated both the main paper and the supplementary material, marking the changes in $\color{red}{\text{red}}$ for readability.
> > >
> > > If there are further improvements, you would like, please let us know.

---

> > > > ### Comment · Reviewer_Knd8 · 2022-08-08
> > > > **Increased score to 7, as promised!**
> > > >
> > > > Thank you to the authors for addressing my concerns.

---

### Official Review · Reviewer_nzwx · 2022-07-10

**Rating:** 6
**Confidence:** 4
**Soundness:** 3 good
**Presentation:** 3 good
**Contribution:** 2 fair

**Summary:**

This paper proposes a dual network architecture extension of PPO. It considers three modifications: a separate neural network for value function and policy, calibration of bias-variance trade-off, and constrained distillation. The method is evaluated on a new Atari-5 environment and compared with a similar baseline Phasic Policy Gradient (PPG).


**Questions:**

How is the combined score in Figure 6 achieved? The performance improvement is visible only in the “Phoenix” environment and overlaps in other environments. Any aggregated metrics, such as the probability of improvement proposed in [1], might give a better understanding. How many seed run was used to generate Figure 6? It is mentioned that “multiple seeds run” in line 175.

The baseline PPG was heavily tested on the Procgen benchmark [2] for sample efficiency. Thus a comparison of the Procgen benchmark seems critical to assess the significance of the proposed method. Therefore, I suggest adding the experiment on Procgen to verify how the proposed method performs on existing and well-established benchmarks in addition to the newly proposed Atari-5.


**Limitations:**

Yes. Limitations and potential societal impacts are discussed.

**Strengths And Weaknesses:**

### Strength:
- Tackle the important problem of improving the policy gradient method, which has wide applicability
- Overall, well-written paper and easy to follow.
- Show some improvement over baseline PPG on the Atari-5 benchmark.

### Weakness (Details in Questions):
- Lack of empirical evidence that demonstrates DNA’s advantage.
- Missing experiment on Procgen benchmarks where the baseline PPG was evaluated.

---

> ### Author Response · Authors · 2022-07-31
> **ProcGen**
>
> Hi,
>
> Thank you very much for your review. We will address your questions and weakness.
>
> 1. Lack of empirical evidence - We agree with this, and other reviewers have made similar comments. Because of this, we have now produced results on two additional benchmarks, MuJoCo and ProcGen, both of which show a significant improvement over PPO.
>
> 2. Procgen - We have now included results on the ProcGen benchmark. Because of the random procedural generation, ProcGen benefits greatly from a large replay buffer, which we explicitly did not include in DNA. This is because, for many tasks, it is not needed (as shown by our Atari and MuJoCo). Our results on ProcGen, which we will include in an updated appendix shortly, puts DNA between PPO and PPG on this benchmark. Outperforming PPG on this benchmark is very difficult, as it was specifically designed to address the unique challenges of the problem. Despite this, DNA can produce results nearing PPG's performance using only four updates (compared to PPGs 8) and without the need for a large replay buffer.
>
> 3. Score - The combined score was generated using a weighted geometric mean. This procedure is described in our accompanying paper Atari-5, but for clarity, we have now included a paragraph explaining the process (and justification) in the paper itself. In terms of performance improvement, DNA outperforms PPO in our experiments on all five games tested. However, not all of these results are to a statistically significant degree. When averaged across all games, the difference becomes quite large. PPG's performance was hampered by poor scores on NameThisGame, Qbert and Pheonix.
>
> 4. Other metrics - This is an interesting idea. We like the idea of using measures outside of the expected score. We have some future work that takes a deep dive into this.
>
> 4. Seeds - We performed either 3 or 5 seeds. Each figure has a note indicating how many seeds were used to produce it.
>
> 5. Procgen - We agree with this point and have now run DNA on the ProcGen benchmark. Results will be uploaded shortly.
>
> We hope that the addition of these supplementary experiments will address your concerns.

---

> > ### Comment · Reviewer_nzwx · 2022-08-08
> > **New experiments improved the paper**
> >
> > I appreciate the authors’ effort in revising the paper and conducting more experiments which overall improved the paper. However, from the empirical performance, it is unclear if DNA is better than PPG. Authors speculate that using a replay buffer helps PPG achieve superior performance in Procgen. However, not using a replay buffer is one of the motivating factors of DNA. I suggest toning down the claim of DNA having a better empirical result than its predecessor, for example, in the Abstracts, and throughout the paper.
> >
> > Overall, empirical evidence is lacking regarding the advantage of DNA over relevant baseline (e.g., PPG). However, I also think that should not be the only criteria to judge the contribution of a paper. Thus, I increase my score from 5 to 6.

---

> > > ### Author Response · Authors · 2022-08-08
> > > **Response**
> > >
> > > Thank you for your review and feedback. We have amended the language in the paper to clarify that our results show DNA outperforming PPG specifically on the Atari-5 benchmark rather than the previous (broader and erroneous) claim that it outperforms it in general. This change will appear in the camera-ready version.

---

### Official Review · Reviewer_2gBU · 2022-07-11

**Rating:** 7
**Confidence:** 4
**Soundness:** 4 excellent
**Presentation:** 3 good
**Contribution:** 3 good

**Summary:**

The paper analyzes various design decisions made by PPO, a popular actor critic policy gradient method, and proposes various fixes to the existing algorithm/architecture to improve the performance. Specifically, they study the impact of the objective noise levels on the policy training and how it interacts with the various design decisions like the network architecture, return estimation, batchsize etc. They use the analysis to propose improvements to the algorithm and the architecture which results in improvements to the policy performance when tested on the Atari-5 benchmark.

**Questions:**

I think both the points in the Weaknesses sections are fixable and would appreciate the authors making amends along those directions.

**Minor Comments**
1. I think the following was a typo : Line 185 : policy -> value

**Limitations:**

Yes.

**Strengths And Weaknesses:**

**Strengths**
1. The paper provides simple, well motivated fixes to PPO/PPG that demonstrate improved performance on the Atari Benchmark!
2. The paper is very well written and easy to follow. I also really liked the motivating example in section 3.1 to demonstrate the impact of noise scale on the architecture.
3. I liked the careful ablations in sections 6.1 and 6.2 analyzing the impact of the specific hyperparameters.
**Weaknesses**
1. Although the paper mentions that they leave the continuous control tasks for future work, I would still have liked some analysis of it in this paper itself to better understand the implications on those settings as well since PPO is widely used there.
2. There have been various other papers in the field analyzing the various aspects of PPO over the past few years. I would have liked a more thorough review of the literature especially along those lines and see this work placed in the context of those papers.

---

> ### Author Response · Authors · 2022-07-31
> **Additional Experiments**
>
> Hi,
>
> Thank you for your review. We agree with the importance of adding experiments on MuJoCo. One of the reasons we had not done this is that PPO already performs quite well on this benchmark but has traditionally performed poorly on Atari. We wanted to address this weakness. However, out of curiosity, we ran some experiments on Mujoco earlier this week and found that DNA is surprisingly strong on this MuJoCo, outperforming PPO on 5/8 of the tasks tested and being of roughly equal performance on the remaining 3.
>
> We have now included these new results in the appendix and will upload a revised copy shortly. Thank you for suggesting this. We are also in the process of adding results on ProcGen, as suggested by another reviewer. We hope these two experimental additions address your concerns and round out the paper nicely.
>
> Regarding the literature review, we have found some additional references that would be useful here and would be happy to modify the literature review to include a broader discussion on PPO. We have quite a long list of papers already, but if you have any specific ones you think should be included, please let us know.

---

> > ### Comment · Reviewer_2gBU · 2022-08-07
> > **Response to Author's comments**
> >
> > I appreciate the authors putting in the effort to perform additional experiments! I also appreciate the author's honesty in reporting Procgen results as well even though they are somewhat worse compared to PPG. Considering these revisions, I would like to bump up my score to 7.

---

### Author Response · Authors · 2022-07-31
**Additional Experiments**

A general theme of the feedback is that the paper would benefit from additional experiments. We wholeheartedly agree. Therefore we have run DNA on the MuJoCo and ProcGen benchmarks. Our results will be uploaded via an update to the supplementary material over the next few days.

The summary of the results is that DNA outperforms PPO in 5/8 of the MuJoCo environments tested and is roughly equal in performance on the other 3 (DNA and PPO both likely score maximum or close to maximum points on these other environments)

For ProcGen, DNA outperforms PPO and underperforms PPG. This was expected as ProcGen's random generation requires a large replay buffer to stabilize training over the 200 variations that must be learned.

We ran these experiments last week and did not have time to fully tune hyperparameters. Despite this, the results still show a clear improvement of DNA over PPO. This puts DNA better on 3/3 benchmarks compared to PPO and better on one, worse in another compared to PPG.

---

> ### Author Response · Authors · 2022-08-02
> **Revision to Appendix**
>
> We have uploaded a revised appendix as part of the supplementary material. This appendix contains two new sections K, and L, detailing our supplementary experiments on the MuJoCo and Procgen benchmarks. We will upload a revised version of the main paper in the next few days that addresses the remaining issues.

---

> ### Author Response · Authors · 2022-08-06
> **Revisions to Paper**
>
> Thank you very much for everyone's kind suggestions. We have made several changes to the paper based on your feedback. A  revised version how now been uploaded with the following modifications.
>
> - Appendix added with MuJoCo results (added to supplementary material only and in the previous update)
> - Appendix added with Procgen results (added to supplementary material only and in the previous update)
> - Fixed typo on line 185 (policy->value) (thanks for spotting this).
> - Added a new section in the literature review, giving a bit more context to PPO and the recent developments.
> - Added an explanation of Atari-5 and why we used it. [section 5, first paragraph]
> - Modified section 4.2 final paragraph to explain why we hypothesised that bootstrapping + bias would be more of a problem than bootstrapping + variance.
> - Made it clearer why our experiments started with a search over training epochs.
> - Included a discussion regarding UTD under future work. (our guess is that the optimal UTD ratio is environment dependant, and that, counterintuitively, high variance environments require fewer epochs, so you don't overfit the noise, but we will leave that for later...).
>
> We had to cut out a few paragraphs to fit some of the new changes into the 9-page limit. However, we believe an additional page is allowed for the document-ready version, so we will restore these paragraphs in the final version.
>
> Again, thank you, everyone, for your reviews and feedback. We enjoyed reading them and believe the paper is much improved as a result.

---

### Meta-Review · Area_Chair_DaEH · 2022-08-30

**Recommendation:** Accept
**Confidence:** Certain

**Metareview:**

The reviewers found this to be a well-executed technical contribution, and all reviewers agree it meets the bar for acceptance.  While this paper does not seem to provide a breakthrough novel insight, it does contribute useful information for the field, and I believe sharing with the community is beneficial.  I recommend accepting this paper.



**Award:**

No

---

### Decision · Program_Chairs · 2022-09-14

Accept